# Advances in Polymeric Micelles: Responsive and Targeting Approaches for Cancer Immunotherapy in the Tumor Microenvironment

**DOI:** 10.3390/pharmaceutics15112622

**Published:** 2023-11-13

**Authors:** Lichun Cheng, Jiankun Yu, Tangna Hao, Wenshuo Wang, Minjie Wei, Guiru Li

**Affiliations:** 1Department of Pharmacy, The Second Hospital of Dalian Medical University, Dalian 116027, China; chenglichun-88@dmu.edu.cn (L.C.); haotangna@dmu.edu.cn (T.H.); wangws@dmu.edu.cn (W.W.); 2School of Pharmacy, China Medical University, Shenyang 110122, China; jkyu@cmu.edu.cn

**Keywords:** polymeric micelles, cancer immunotherapy, tumor microenvironment, responsive delivery systems, targeted drug delivery

## Abstract

In recent years, to treat a diverse array of cancer forms, considerable advancements have been achieved in the field of cancer immunotherapies. However, these therapies encounter multiple challenges in clinical practice, such as high immune-mediated toxicity, insufficient accumulation in cancer tissues, and undesired off-target reactions. To tackle these limitations and enhance bioavailability, polymer micelles present potential solutions by enabling precise drug delivery to the target site, thus amplifying the effectiveness of immunotherapy. This review article offers an extensive survey of recent progress in cancer immunotherapy strategies utilizing micelles. These strategies include responsive and remodeling approaches to the tumor microenvironment (TME), modulation of immunosuppressive cells within the TME, enhancement of immune checkpoint inhibitors, utilization of cancer vaccine platforms, modulation of antigen presentation, manipulation of engineered T cells, and targeting other components of the TME. Subsequently, we delve into the present state and constraints linked to the clinical utilization of polymeric micelles. Collectively, polymer micelles demonstrate excellent prospects in tumor immunotherapy by effectively addressing the challenges associated with conventional cancer immunotherapies.

## 1. Introduction

Cancer remains a major threat to human health. Cancer cells evade immune detection through various mechanisms, including immune checkpoint molecules and cellular systems that promote immune suppression [1,2]. Cancer cells and the tumor microenvironment (TME) can express various signaling molecules, such as programmed death receptor 1 (PD-1), programmed death ligand 1 (PD-L1), cytotoxic T lymphocyte antigen-4 (CTLA-4), and indoleamine 2,3 oxygenase (IDO), which can help them evade immune surveillance and induce immune cell anergy [3,4,5]. Cancer immunotherapy has become an effective treatment following surgery, chemotherapy and radiotherapy [6] Cancer immunotherapy is a concept of utilizing intrinsic mechanism of host immune system to distinguish and destroy malignant cells, involving passive immunotherapy, active immunotherapy, and immune checkpoint blockade [7,8]. At present, cancer vaccines, immune-activating cytokines, adoptive cell transfer therapy (ACT), and immune checkpoint inhibitors (ICIs) are the most popular and available cancer immunotherapy modalities so far [9,10,11]. One of the earliest immunotherapies is that use high dose interleukin-2 to activates T-cells [12]. ICIs and ACT have been proved to be effective for various malignant tumors [13,14,15]. It has been confirmed that cancer immunotherapy can prevent the metastasis, recurrence and reversal of multidrug resistance of tumor cells [16,17,18]. So far, cancer immunotherapy has been proven to be effective for head and neck cancer, lung carcinoma, leukemia, breast carcinoma, ovarian carcinoma, renal carcinoma, and bladder tumor [19,20,21,22,23].

However, immunotherapies have side effects and are not as effective due to problems such as tumor heterogeneity, short half-life, and unintended toxic effects. For example, ICIs like PD-L1/PD-1 have been employed in diverse tumor immunotherapies; however, they exhibit low response rates in a subset of cancer patients [24,25,26,27,28]. Nanotechnology can address pharmacokinetic deficiencies by enhancing permeation and retention (EPR) using synthetic nanocarriers with specific features such as dimensions, morphology, surface ligands, loading techniques, zeta potential, water affinity, flexibility, and biocompatibility [29,30]. These nanocarriers can serve as suitable carriers for immunotherapeutic molecules, extending their biological half-life, protecting them during circulation, penetrating barriers, or having targeting effects, and facilitating precise payload release within the TME while minimizing biotoxicity [31].

Polymeric micelles consist of amphiphilic copolymers, featuring an external hydrophilic shell and an internal lipophilic core. These spherical colloidal particles typically exhibit sizes ranging from 10 to 100 nm. They excel in encapsulating and ferrying poorly water-soluble drugs, mitigating micelle biofouling to extend circulation half-life, facilitating sustained drug release at optimal concentration levels, and offering the possibility of additional functionalization with targeting ligands to achieve precise delivery [32,33]. Even at exceedingly low polymer concentrations, the structural stability of polymer micelles retains its utmost importance. This stability is attributed to their low critical micelle concentration, which enhances their circulation in the bloodstream compared to surfactant micelles [34]. The properties of polymeric micelles, such as their straightforward formulation, uncomplicated structure, drug solubilization capabilities, enhanced biocompatibility, pharmacokinetics, biodistribution improvements, and the potential for further customization, make them widely used in anti-tumor research [35]. Ongoing clinical trials involving polymer micelles loaded with anticancer drugs, for instance, paclitaxel loaded micelles Genexol^®^-PM and NK105, docetaxel-micelles Nanoxel^®^M and BIND-014, doxorubicin-micelles NK911 and SP1049C [33,36]. As cancer immunotherapy advances, research on the use of polymeric micelles in this field has become very extensive. This review will concentrate on the responsive application of polymeric micelles, their role in remodeling the TME, and their applications in cancer immunotherapy, shown in Figure 1.

## 2. Responsive and Remodeling of the TME

TME is a highly intricate milieu comprising normal and cancerous tissue-resident cells, immune cells, vascular cells, fibroblasts and components of the extracellular matrix [37,38]. Unfortunately, the immunosuppressive TME limits the efficacy of immunotherapy [39]. In contrast to healthy tissue, the TME exhibits an acidic pH, upregulation of certain enzymes, increased redox potential, ROS, and hypoxia [40,41]. Polymeric micelles, as a nanocarrier that responds to the tumor microenvironment, could enhance tumor immunotherapy, and are shown in Figure 2.

pH-responsive polymeric micelles: TME is often characterized by acidity, primarily due to the abnormal vasculature and hypoxia within the tumor. Tumor cells heavily rely on glycolysis, an oxygen-independent metabolic pathway, to generate energy. This metabolic process results in increased production and excretion of H^+^ ions, leading to a decrease in the extracellular pH of the TME to a range of 6.5−6.9, in contrast to the normal extracellular pH of 7.2−7.4 [42]. pH-responsive polymer micelles can undergo self-assembly into micelles at ambient pH values above their specific pKa, which is determined by functional groups within the polymer chains, such as pyridine, L-histidine, and tertiary amine moieties. This property allows the micelles to respond to the acidic TME and undergo structural changes, enabling targeted drug delivery to the tumor site [43,44,45]. Numerous investigations have reported the use of pH-responsive micelles for various applications, including vaccine platforms. For instance, histidine-modified stearic acid-grafted chitosan, dilauroyl phosphatidylcholine and deoxycholic acid, diblock copolymer, comprising carboxyl-modified poly(2-ethyl-2-oxazoline), and poly(d, l-lactide) have been utilized as pH-responsive micelles for loading ovalbumin as vaccine platforms. These pH-responsive micelles have shown their capability in triggering cellular immunity, resulting in improved antitumor immune reactions [46,47,48]. The precise pH-responsive co-delivery of imiquimod and doxorubicin using micelles exhibited not only elevated tumor accumulation and decreased side effects but also a marked rise in the quantity of mature dendritic cells (DCs), activated cytotoxic T cells, and M1-like macrophage polarization [49].

Redox-responsive polymeric micelles: Tumor cells regulate their reducing environment through the presence of NADPH and glutathione (GSH). GSH regulates the microenvironment and exhibits elevated levels in tumor tissue compared to normal tissue [50]. Redox-sensitive micelles release drugs rapidly and deform in the reducing intracellular environment due to high GSH concentration in tumors [51]. The cRGD-functionalized and reduction-sensitive polymeric micellar mertansine prodrug (cRGD-MMP), characterized by enhanced stability, drug loading, and the ability to target αvβ3, presents an appealing substitute for antibody-maytansinoid conjugates in the treatment of malignant tumors [52]. A redox-sensitive polymer, POEG-co-PVDGEM, which incorporates gemcitabine (GEM), was employed as a compact nanocarrier to simultaneously transport hydrophilic GEM and hydrophobic paclitaxel. This approach proved to be an efficient strategy for improved tumor penetration and enhanced antitumor immune responses [53]. Another co-delivery system used a polymeric prodrug carrier, incorporating an immune checkpoint inhibitor (NLG919) and the chemotherapeutic agent doxorubicin. This strategy markedly suppressed tumor growth in mice with 4T1.2 tumors and fostered a more immune-responsive tumor microenvironment [54]. Chitosan-coated hyaluronic acid micelles that are responsive to pH and redox conditions were designed to improve the targeted delivery of doxorubicin and siPD-L1 to tumors. This method represents a promising approach to synergize chemotherapy and siRNA-based immunotherapy for enhanced effectiveness [55].

ROS-responsive polymeric micelles: Reactive Oxygen Species (ROS) have a close association with cancer development, with cancer cells exhibiting ROS levels up to 100 times greater than those in normal cells [56,57]. ROS-responsive vesicles for cancer targeting have sparked scientific interest due to their outstanding biocompatibility, rapid response time, and the ability to load a significant amount of drug [58]. Galactose-functionalized zinc protoporphyrin IX grafted poly(l-lysine)-b-poly(ethylene glycol) polypeptide micelles loaded with Poly I:C efficiently repolarized TAMs to M1 macrophages via ROS generation [59]. A potential platform technology for cancer immunotherapies involves a polyion complex consisting of a cationic polyamine-poly-polyamine triblock copolymer with ROS-scavenging side chains, an anionic poly(acrylic acid), and a protein [60]. Recently, I-P@NPs@M, is documented inhibits lung metastasis by chemotherapy, photodynamic therapy, and immunotherapy gathering. The chlorin e6 within I-P@NPs@M is capable of converting a 650 nm laser into ROS, which induces the transformation of spherical micelles into nanofibers, enhancing their retention within the tumor region. As a result, the elongated nanofibers can stably release drugs over time [61]. Another sequential-targeting micelle was developed, consisting of a cationic amphiphilic copolymer core loaded with chlorin e6, and a pH-sensitive charge-altering layer designed for tumor targeting. This layer is derived from 2,3-dimethylmaleic anhydride-modified Biotin-PEG_4000_-NH_2_ through electrostatic interactions. This innovative approach utilizes the sensitivity of cellular organelles to ROS and proves to be an effective method for promoting efficient photodynamic therapy and immune response [62].

Enzyme-responsive polymeric micelles: The overexpression of enzymes by tumor cells and the TME represents another significant trigger for the responsiveness of TME-targeted micelles. Matrix Metalloproteinases (MMPs), a class of calcium-dependent endopeptidases, utilize a trio of amino acids to serve as ligands for coordinating zinc ions, enabling them to perform their catalytic functions. Dysregulation of MMPs can impact various physiological processes such as morphogenesis, tissue remodeling, embryonic development, and the control of cellular growth and apoptosis [63]. Given that MMPs play a crucial role at all stages of metastasis, this study delves into their involvement in cancer, focusing on the distribution and mobility of MMPs within cells and tumors to exploit their potential for cancer-targeting applications [64]. A dual-sensitive micelle system, responsive to both MMPs and pH, has been developed for co-delivering anti-PD-1 antibodies and paclitaxel, resulting in a high response rate to immune checkpoint inhibitors (ICIs) and an effective chemoimmunotherapy that leverages the TME’s sensitivity [65]. Similarly, there is promising potential in a dual-sensitive micelle–liposome system, which responds to both enzymes and pH. This system co-delivers paclitaxel (PTX) and the PD-1/PD-L1 inhibitor HY19991 for the treatment of breast cancer [66].

Remodeling TME: Cancer cells employ diverse mechanisms to restructure the TME, thereby evading immune surveillance and fostering tumor proliferation and metastasis. The TME encompasses an intricate interplay of cancer cells, the extracellular matrix (ECM), stromal cells, and immune cells. Throughout the course of cancer development and progression, the ECM within the TME undergoes extensive remodeling. This remodeling can exert inhibitory effects on immune responses, frequently resulting in inadequate or less-than-optimal tumor responses to immunotherapy in most instances. Strategies in immunotherapy encompass a range of approaches such as adoptive cell therapy (ACT), vaccines, and immunomodulatory antibodies, all aimed at reshaping the immunosuppressive milieu of the tumor microenvironment towards an environment conducive to immune support [67]. Many researchers have attempted to enhance drug-loaded nanocarrier delivery efficiency by modifying the ECM within the TME [68,69]. Chanyoung Song and colleagues investigated an innovative approach using an injectable immunomodulatory multidomain nanogel (iGel), which effectively addresses this challenge by converting the pro-tumoral TME into anti-tumoral immune-friendly niches. The iGel is formed through electrostatic interactions between negatively charged non-concentric multi-nanodomain vesicles and positively charged nanoliposomes, both loaded with immunomodulatory agents. The gel structure can be temporarily disassembled under shear force during syringe injection and reassembled once shear force is removed at the treatment site. The iGel serves as an immunotherapeutic platform capable of reshaping immunosuppressive TMEs and synergizing with checkpoint therapies in cancer immunotherapy, all while reducing systemic toxicity [70]. Furthermore, micelle-like PLGA-PEG-anisamide nanocarriers loaded with cisplatin and rapamycin, when co-encapsulated with rapamycin, exhibited a 3.5-fold improvement in rapamycin loading efficiency. Co-delivery of these micelles resulted in reduced numbers of tumor-associated fibroblasts and decreased levels of collagen expression within xenograft tumors. These micelles demonstrated antiangiogenic effects, normalization of tumor blood vessels, and enhanced penetration in A375-luc human melanoma [71]. Additionally, SUNb-PM, an engineered polymeric micelle delivery system, exhibited synergistic effects when used in conjunction with vaccine therapy in an advanced mouse melanoma model. SUNb-PM not only facilitated the transformation of TAFs, collagen, and tumor vasculature but also induced tumor cell apoptosis, reducing tumor immune evasion by inhibiting the Stat3 and AKT signaling pathways [72].

## 3. Regulating Immune-Suppressing Cells in the TME

Tumor-associated immunosuppressive cells encompass tumor-associated macrophages (TAMs), myeloid-derived suppressor cells (MDSCs), and Regulatory T Cells (Tregs) [73]. TAMs constitute the most prevalent immune cell population within the TME [74]. Indeed, TAMs play a pivotal role as orchestrators of cancer-associated inflammation and are increasingly recognized as essential contributors to tumor advancement while simultaneously impeding antitumor immune reactions [75]. Diverse therapeutic approaches targeting TAMs have been devised, encompassing measures to prevent macrophage recruitment to the tumor, eliminate TAMs directly, reprogram TAMs from their pro-tumoral M_2_-like state to an anti-tumoral M_1_-like state, and utilize TAMs for the delivery of therapeutic payloads [76]. In this review, we discuss published studies that have utilized micelles to modulate TAMs, including depleting TAMs, reprogramming TAMs, and targeting TAMs (Table 1).

Tregs which are one of the tumor-associated immunosuppressive cells, have been observed to facilitate tumor cell growth and advancement. To address this issue, a multifunctional immunostimulatory polymeric prodrug carrier, PEG2k-Fmoc-1-MT, was created to deliver 1-methyl tryptophan and the chemotherapeutic doxorubicin. This carrier effectively promotes the activation of CD4^+^ and CD8^+^ T cells, while concurrently decreasing the expression of Tregs, thus enhancing the efficacy of immunochemotherapy for breast cancer [86]. Another effective approach involves the use of hybrid micelles (SK/siIDO1-HMs) for the delivery of shikonin and siRNA targeting IDO-1 knockdown, which have shown promising results in suppressing Tregs in the tumor microenvironment [87].

In addition, MDSCs are among the crucial immunosuppressive cells within the tumor microenvironment, fostering the growth and advancement of tumor cells. In order to specifically target and counteract MDSCs, a dual-pH-sensitive conjugated micelle system (PAH/RGX-104@PDM/PTX) has been created for the targeted delivery of the liver-X nuclear receptor agonist RGX-104 and paclitaxel to the perivascular region and tumor cells. This system effectively eliminates MDSCs and enhances the infiltration of cytotoxic T lymphocytes, thus exerting potent antitumor effects [88]. Another promising approach is the use of dual-functional micelles, such as Dox/PEG-Fmoc-NLG, which have the ability to decrease both MDSCs and Tregs. These micelles hold great potential for immunochemotherapy in lymphoma treatment [89]. Sun et al. developed a TME charge reversal system, HA/pIL-12/DOX-PMet, which synergistically enhances NK cell and tumor-infiltrating cytotoxic T lymphocyte activity, shifts M_2_ macrophage polarization to an activated antitumor M_1_ phenotype, reduces Tregs, and elevates cytokine expression (IL-12, TNF-α, IFN-γ). These combined effects improve antitumor and anti-metastatic outcomes in a 4T1 breast cancer lung metastasis mouse model [90].

## 4. Enhancing Immune Checkpoint Inhibitors (ICIs)

ICIs have demonstrated tremendous potential in leveraging the immune system to combat cancer. Some prominent examples of ICIs include PD-1, PD-L1, IDO, CTLA-4, CD47, and CD40, among others.

PD-1, part of the CD28 immunoglobulin superfamily, is predominantly found on the surface of activated T cells, B cells, and various other immune cells, including NK cells and myeloid cells. Its immunosuppressive function relies on the interaction with its ligands, PD-L1 and PD-L2, which also function as immune checkpoints. PD-L1 and PD-L2 are mainly found on tumor cells, further contributing to the immunosuppressive microenvironment [91]. Despite their clinical success, there are limitations to the use of PD-1/PD-L1 inhibitors, such as predicting patient response to these inhibitors and immune escape remain challenges [92]. Polymer micelles can enhance the efficacy of antibodies; however, their broad adoption is impeded by factors such as high cost, instability, and the potential risk of autoimmune diseases. Additionally, polymer micelles have shown promise in augmenting patient response to checkpoint immunotherapy while simultaneously reducing treatment complications. The utilization of siRNA@PPDS micelles in the combined therapy targeting PD-L1-KD and HDACIs represents a potential and effective approach to overcome immune checkpoint inhibitor resistance and provide a promising treatment option for inhibiting tumor growth [93]. Cancer immunotherapy through photodynamic therapy (PDT) has been established utilizing a versatile micelleplex system. This system integrates an acid-activatable cationic micelle, small interfering RNA, and photosensitizer [94]. In addition, a hyaluronic acid (HA) linked to chlorin e6 (Ce6) forms a HA-Ce6 conjugate (HC). Encapsulation of a small-molecule inhibitor, BMS 202 (BMS), within BMS/HC micelles improves the PD-1/PD-L1 blocking efficacy and facilitates efficient photoimmunotherapy [95]. Researchers have investigated the co-delivery of paclitaxel and anti-PD-1 antibody using micelles to enhance tumor chemoimmunotherapy [65,96]. In one study, a dual-responsive carboxymethyl chitosan micelle functionalized with a targeting peptide GE11 was developed. This micelle allowed the concurrent transport of doxorubicin and PD-L1 siRNA. The micelle effectively inhibited immune escape and significantly improved the anti-tumor immune response, resulting in suppressed tumor growth [97]. In another study, a cocktail strategy involving paclitaxel, thioridazine, and the inhibitor HY19991 of PD-1/PD-L1 was incorporated into the micelle for the treatment of breast cancer. This micelle formulation led to a decreased proportion of cancer stem cells and increased T cell infiltration within tumor tissues [66]. Seungpyo Hong and his research team have conducted thorough investigations into the utilization of nanopolymer materials for cancer therapy. They employed hyperbranched, multivalent poly(amidoamine) dendrimers to prepare dendrimer-ICI conjugates, which improved the PD-L1 blockade effect through the binding affinity of the inhibitor to the target proteins [98]. They found that poly(amidoamine) (PAMAM) exhibited the greatest proficiency in capturing exosomes cultured in human serum [99]. Recently, they harnessed PAMAM to engineer an innovative hybrid NP system that melds the favorable biological traits of exosomes with gene delivery, leading to a substantial reduction in PD-L1 expression (3.8-fold more than dendrimers alone, *p* < 0.05). Their findings illustrate that both exosomes and dendrimers propose a novel nanomicelles design strategy [100].

CTLA-4 is upregulated upon T cell activation, and its protein sequences of CTLA-4 share high homology with CD28. CTLA-4 possesses two ligands, namely CD80 (B7-1) and CD86 (B7-2) [101]. Monoclonal antibodies that target CTLA-4, such as Ipilimumab and Tremelimumab, have been used in clinical treatments [102]. In a combination approach, a polymer micelle was employed to co-encapsulate the PARP inhibitor Niraparib and the PI3K inhibitor HS-173, alongside anti-CTLA-4 immunotherapy and X-ray irradiation [103].

IDO is an enzyme responsible for catalyzing the conversion of the vital amino acid l-tryptophan into kynurenine, leading to local depletion of tryptophan. This depletion has been demonstrated to trigger anergy and apoptosis in effector T cells [104]. Moreover, IDO is frequently overexpressed in various types of cancer and has been implicated in tumor-mediated immunosuppression [105]. IDO inhibitors include NLG919, PF-06840003, Norharmane, and 1-methyl-DL-tryptophan (1-MT) [106]. Given that IDO is expressed in numerous immunotherapy-resistant cancers, one suggested approach involves constructing PEG micelles effectively inhibiting IDO with 1-MT, such as INCB024360 [107]. NLG919 an IDO-1 inhibitor, has entered clinical trials, demonstrating high IDO selectivity with an EC50 of 75 nM.s [108]. Nonetheless, NLG919 faces a significant hurdle in its clinical application due to its poor water solubility, which hinders therapeutic delivery [109]. Co-delivery IDO inhibit NLG919 and chemotherapy drug (paclitaxel, doxorubicin) micelles a significantly improved anticancer response [54,110,111]. A study revealed that the administration of gefitinib through PEG5k-Fmoc-NLG919 micelles with immunostimulatory properties enhanced the susceptibility of lung cancer cells to gefitinib [112]. The combination of NLG919/IR780 micelles with immunotherapy and photothermal therapy (PTT) not only allows for effective tumor margin suppression through PTT, but also enhances the immune response to inhibit distal tumors [113].

## 5. Engineering Targeting Polymeric Micelles as Cancer Vaccine Platforms

A cancer vaccine is a method that utilizes tumor antigens, immunocytes, or other immune molecules to stimulate the immune system and trigger an immune response [114]. The cytotoxic T-lymphocyte (CTL) response is a crucial immune response in cancer vaccines. Re-activating CTL response within tumor tissues through checkpoint blockade has shown significant success in tumor immunotherapy. PEG-PE micelles, serving as cancer vaccine platforms, enable co-delivery of tumor antigens and monophosphory lipid A adjuvant, leading to a remarkable increase in CTL response [115]. Based on the origin of cancer vaccines, they can be categorized into several types, including tumor cell vaccines, dendritic cell (DC) vaccines, lymph node (LN) vaccines, peptide vaccines, gene vaccines, and more [116,117]. However, there are obstacles in the development of cancer vaccines, such as weak immunogenicity, short half-life, susceptibility to immune tolerance, and major histocompatibility complex (MHC) restrictions [118]. Polymer micelles offer a solution to these challenges and have emerged as excellent platforms for tumor vaccines. By modifying the properties of polymer micelles, such as pH responsiveness and long circulation, they can be utilized as carriers for tumor vaccines, enhancing the effectiveness of tumor immunotherapy [46,48,119,120]. Several studies have reported that the carrier materials of micelles can serve as immune adjuvants. For instance, cholesteryl PADRE-EGFRvIII epitope-conjugated lipopeptide self-assembled micelles have been investigated as a potential self-adjuvant vaccine. Additionally, M-COSA micelles have been used to achieve targeted co-delivery of antigen ovalbumin and plasmid DNA encoding CCR7 [121,122]. Tumor-targeted micelle vaccines have demonstrated the ability to improve therapy for advanced melanoma by modifying the tumor microenvironment [72]. Cell-penetrating peptides (CPPs), typically comprising 4–30 amino acids, possess the capability to permeate cell membranes without inducing substantial toxicity. This property renders them a straightforward and practical choice for intracellular delivery. CPP-conjugated immune modulators can enhance antitumor immune responses or anti-inflammatory effects, offering potential applications in the regulation of allergies and autoimmunity [123]. A nanovaccine, engineered by encapsulating OVA modified with CPPs, amplifies the cytosolic processing of antigens and subsequently augments antigen cross-presentation through MHC-I molecules, thereby eliciting cytotoxic CD8^+^ T cell responses [124]. Moreover, by utilizing a small library of antigenic peptides that undergo antigen folding, the loading efficiency into PC7A micelles is enhanced, resulting in increased antitumor efficacy against melanoma [125]. Here, we summarize the current polymeric micelles as cancer vaccine platform or cancer immunotherapy (Table 2). These polymer micelles offer a range of benefits, including enhanced immunogenicity, increased uptake by antigen-presenting cells (APCs), and the stabilization of the antigen.

## 6. Modulating Antigen Presentation

Antigen-presenting cells (APCs), including DCs and macrophages, play a crucial role in immunomodulation. They play a crucial role in antigen presentation, involving the uptake, processing, and presentation of foreign antigens along with MHC class II molecules to T cells. Efficiently delivering tumor antigens and immunostimulatory adjuvants to lymph nodes is vital for APC maturation and activation, ultimately leading to the induction of adaptive anti-tumor immunity. The specialized vaccine with a cytosol delivery micelle cascade led to an increased rate of MHC I molecule combination and improved antigen cross-presentation efficiency, which was further validated by elevated quantities of CD3^+^CD8^+^ T cells, CD3^+^CD8^+^25D11.6^+^ T cells, and post-subcutaneous secretion of IL-2 and IFN-γ [126]. In another study, Nak Won Kim explored the use of an amphiphilic triblock copolymer-based dissolving microneedle system to deliver a receptor 7/8 agonist. The use of microneedles that included a tumor model antigen (OVA) and R848 applied to the skin of EG7-OVA tumor-bearing mice produced a substantial antigen-specific humoral and cellular immune response, ultimately leading to remarkable antitumor effects [134].

DCs are the most potent APCs play a vital role in initiating and modulating the tumor immune response [135]. As described above, polymer micelles serve as effective delivery platforms for vaccines and can be specifically targeted to DCs through mannose modification, thereby enhancing tumor immunity [129,130]. Various polymer materials have been explored for the development of micelles that can specifically target DCs. These include mannosylated HPMA-LMA block copolymers, HPMA modified with mannose or trimannose carbohydrates, laurylmethacrylate-co-hymecromone-methacrylate, α-galactosylceramide, and the hyperbranched polymer Boltorn H40, among others [136,137,138,139]. Activator of transcription 3 (STAT3) plays a significant role in the progression of cancer cells and cancer-associated DCs. The self-associating polymer, poly(ethylene oxide)-block-poly(α-carboxylate-ε-caprolactone), loaded with the inhibitor JSI-124 of STAT3, forms self-assembled polymeric micelles. The results demonstrate that JSI-124 micelles passively target melanoma tumor cells and tumor-associated DCs, leading to the modulation of the immunosuppressive microenvironment [140]. Co-delivery of imiquimod and antigen-encoding plasmid DNA using polymer nanocarriers synergistically enhances immunity, showing potential for gene-based vaccine approaches [141]. Chenxi Li explored the application of an amphiphilic diblock copolymer, poly(2-ethyl-2-oxazoline)-poly(d,l-lactide), in combination with carboxyl-terminated Pluronic F127 to create mixed micelles for the co-delivery of ovalbumin antigen and Toll-like receptor-7 agonist CL264 to lymph node-resident DCs. These mixed micelles elicited robust in vivo immune responses, including antigen-specific T-cell activation, antigen-specific IgG antibody production, and cytotoxic T-lymphocyte responses [142].

## 7. Modulating Engineered T Cells

Chimeric antigen receptor (CAR)—and T cell receptor (TCR)-modified T cells have arisen as a hopeful avenue for adoptive cell therapy, utilizing artificial receptors to target advanced cancer forms. CAR-T cells can directly identify tumor antigens without relying on the major histocompatibility complex. This therapy has shown success in reducing remission rates by up to 80% for hematologic cancers, notably acute lymphoblastic leukemia and non-Hodgkin lymphomas like large B-cell lymphoma. Initial trials of CAR T cells focused on B-cell malignancies, targeting CD19 or CD20 antigens. Recently, anti-CD19 CAR therapy (UCART19) has demonstrated efficacy in relapsed/refractory hematologic cancer [143,144]. Studies by Kristen M. Hege have shown the potential of CAR-T cells targeting tumor-associated glycoprotein (TAG)-72 in treating solid tumors like colorectal cancer [145]. Despite the remarkable achievements of CAR-T technology in treating acute lymphocytic leukemia and non-Hodgkin’s lymphoma, there remain challenges and limitations [146]. The treatment process is complex, involving extraction and in vitro amplification of T lymphocytes from patients, typically taking 2 to 3 weeks before reinfusion [147]. Moreover, CAR-T cell therapy faces obstacles in treating solid tumors, and cytokine-release syndrome is a unique acute toxicity associated with this therapy [148].

Micelles can overcome barriers in CAR T cell therapy by efficiently binding to peripherally circulating T cells and exhibiting high distribution in the bone marrow, lymph nodes, and spleen. Altered micelles can target solid tumors specifically, boosting CAR T cell treatment effectiveness. To address traditional CAR delivery system issues, a three-part amphiphilic co-polymer, mPEG-bPEI-PEBP, is employed to encapsulate DNA plasmids. This approach solves problems like inadequate biosafety, limited loading capacity, and reduced transfection efficiency in traditional CAR delivery systems [149].

## 8. Targeting Other Components of the TME with Micelles

The TME comprises diverse elements, encompassing tumor parenchymal cells, stromal cells, immune cells, ECM, lymphatic vessels, and blood vessels [150]. Recent studies have focused on utilizing micelles to respond to and remodel the TME, as well as modulate immunosuppressive cells within the TME. Among these components, cancer/tumor -associated fibroblasts (CAFs/TAFs) assume a pivotal role within the tumor stroma. They engage with cancer cells, fostering tumor advancement and progression. The densely packed cellular structure and elevated expression of ECM components by cancer cells and CAFs can hinder drug diffusion, thereby curtailing therapeutic effectiveness. Moreover, the presence of ECM and CAFs can contribute to drug resistance, as cancer cells with high ECM expression and interaction with CAFs exhibit increased resistance to chemotherapy compared to other tumor cells [151]. To overcome these challenges, harnessing the distinctive attributes of micellar drug delivery systems can enable precise drug targeting to tumor cells and improve the efficacy of chemotherapy by bypassing drug resistance mediated by the ECM and CAFs.

Targeting ECM and CAFs/TAFs: ECM is a fundamental component of the TME that acts bidirectionally, both affecting and being affected by tumor cells. In desmoplastic tumors, CAFs/TAFs and the resultant pathological tumor stroma significantly hinder the accessibility and responsiveness of tumor cells to anti-tumor treatments. Remodeling and targeting ECM and CAFs of the TME can enhance cancer therapy. Angiotensin II type I receptor (AT_1_R) is a member of the G protein-coupled receptor superfamily and is overexpressed on both CAFs and tumor cells, including those found in breast tumors and pancreatic duct adenocarcinoma [152,153]. Telmisartan, functioning as an angiotensin II Type I receptor blocker (ARB), exhibits the highest affinity to AT1R among ARBs due to its distinctive “delta lock” molecular structure [154]. There have been several reports on the use of telmisartan to target the ECM by inducing apoptosis in CAFs and reprogramming the TME. For instance, glycolipid-based polymeric micelles were engineered to encapsulate telmisartan and doxorubicin. Another approach involved telmisartan-grafted glycolipid micelles in conjunction with doxorubicin to reprogram the TME, making internal breast cells more vulnerable [155,156]. As previously mentioned, the mannose-modified lipid calcium phosphate nano-micelles-based Trp2 vaccine has been shown to remodel TAFs, blood vessels, and collagen in melanoma therapy [72]. Chao Teng developed a polymeric micelle that responds to fibroblast activation protein-α (FAP-α). This micelle consists of a CD44-targeting outer layer and a polyethylene glycol (PEG) coating that can be cleaved by FAP-α. The FAP-α-responsive polymeric micelle exhibited significant anticancer effects by inducing apoptosis in CAFs and reducing collagen levels within tumor tissues [157]. Moreover, it successfully co-delivered the anti-CAFs agent tranilast and the antitumor agent docetaxel within the micelle, disrupting the communication between tumor cells and stromal cells. This led to improvements in the TME and enhanced antiproliferative effects [158]. In another study, the incorporation of cyclopamine, a sonic hedgehog inhibitor, with paclitaxel in a polymeric micelle resulted in stromal remodeling and enhanced pancreatic cancer therapy [159]. Furthermore, employing a mixed polymeric micelle stabilized with lecithin and loaded with mPEGylated docetaxel facilitated precise tumor targeting and the specific recognition of antigens unique to TAFs. This approach resulted in superior tumor growth inhibition compared to Tynen^®^ while causing fewer adverse effects [160]. Collectively, these studies demonstrate the potential of using micelles for co-delivering CAFs inhibitors and chemotherapeutic drugs to reprogram the TME and enhance the efficacy of anti-tumor treatments.

Targeting tumor neovascularization: Tumor neovascularization, driven by the process of angiogenesis, plays a critical role in supplying nutrients and oxygen to support the growth of neoplastic cells within a tumor. Angiogenesis involves a complex series of events, including the activation and growth of neovascular endothelial cells, modifications to the extracellular matrix, changes in vascular permeability, and the development of new blood vessels. Additionally, tumor neovascularization can significantly impact the TME. Targeting tumor neoangiogenesis and impeding its development is considered a potential therapeutic avenue in cancer treatment. Vascular endothelial growth factor (VEGF) plays a pivotal role in regulating tumor neovascularization, and its inhibition can effectively curb tumor angiogenesis. One method involves the concurrent delivery of siVEGF and PTX, achieved through apelin (Ap)-modified copolymeric micelles and folate-PEG-PHIS micelles. This approach results in substantial suppression of neovascularization via VEGF gene silencing [161,162]. Moreover, integrin αvβ_3_ is a cell adhesion molecule abundantly present on the surface of neovascular endothelial cells. Yupeng Liu developed c(RGDfk)-modified glycolipid-like micelles encapsulating incorporating indocyanine green, enabling dual-targeting of integrin αvβ_3_ on neovascular endothelial cells and glioblastoma [163]. A d-peptide ligand is a ligand that specifically binds to integrins found in abundance on glioma cells and tumor neovasculature. When incorporated into modified micelles loaded with doxorubicin, it significantly enhances the targeting efficiency for glioma [164]. Numerous Chinese herbal medicines have exhibited inhibitory effects on retinoblastoma growth in various research studies, although the underlying mechanism remains poorly understood. Incorporating natural components with chemotherapy drugs in micelles has shown promising results in increasing the anti-tumor effect by inhibiting tumor neovascularization. Some notable examples of these components include curcumin, celastrol, triptolide (LA67), luteolin, and ursolic acid [165,166,167,168,169]. The celastrol nanomicelles effectively inhibited hypoxia-induced VEGF and HIF-1α leading to the suppression of retinoblastoma growth and angiogenesis. Additionally, luteolin MPEG-PCL micelles induced apoptosis by down-regulating Pro-caspase9 and Bcl-2 while up-regulating cleaved-caspase9 and Bax [166,168].

## 9. Clinical Trials and Application Status of Polymer Micelles

Through extensive laboratory research, numerous anti-tumor micellar formulations have progressed to the clinical trial phase (Table 3) [170]. Certain chemotherapy drugs encapsulated within micelles are currently accessible in the market, which includes products like NK105, NC-6004, and Genexol-PM^®^ [171]. However, polymer micelles as a platform for tumor immunotherapy are currently in the laboratory-based research stage. It is anticipated that in the near future, they will advance to the clinical trial phase.

## 10. Limitations of Polymeric Micelles Clinical Application

Polymeric micelles offer the potential to be applied in cancer immunotherapy through multiple avenues, including enhancing the delivery of immunostimulatory agents, improving the pharmacokinetics and biodistribution of immune-modulating drugs, and enhancing the effectiveness of cancer vaccines. While polymeric micelles have shown promise in cancer immunotherapy, there are several limitations that need to be addressed to maximize their potential. Firstly, limited drug loading capacity: Polymeric micelles have limited drug loading capacity, particularly for hydrophilic drugs. This limitation can restrict the quantity of drug delivered to the target location, potentially diminishing its effectiveness. Secondly, short circulation time: Polymeric micelles can be rapidly cleared from the bloodstream, particularly in the presence of serum proteins [172]. This can limit the amount of time the drug is available to exert its therapeutic effect. Thirdly, immune response: Polymeric micelles may elicit an immune response, leading to clearance and reduced efficacy. The last but not least, clinical translation: although polymeric micelles have displayed potential in preclinical investigations, their translation to the clinic is still limited by issues such as manufacturing scalability, regulatory hurdles, and cost-effectiveness [173,174].

## 11. Conclusions and Future Applications of Polymeric Micelles

The future research directions for active-targeting polymeric micelles in cancer immunotherapy can be explored in the following aspects: Firstly, researchers will continue to explore more efficient methods of encapsulating immunotherapeutic agents into polymeric micelles, such as exploring new materials and optimizing their physicochemical properties. Secondly, efforts will be made to enhance the targeting efficiency of polymeric micelles to cancer cells and tissues, either through surface modification or active targeting strategies. For example, the design and engineering of CPPs may facilitate the secure transportation of therapeutic compounds through biological barriers [175]. Thirdly, the development of multifunctional polymeric micelles capable of concurrently delivering various types of immunotherapeutic agents simultaneously may become an important research direction. Fourthly, there is a need to further optimize the release profile of immunotherapeutic agents from polymeric micelles, such as achieving sustained release and targeted release under specific conditions. Finally, more in-depth studies on the safety and biocompatibility of polymeric micelles in vivo are necessary to ensure their clinical translation.

In conclusion, while polymeric micelles have enormous potential in tumor immunotherapy, their application needs further improvement. This includes enhancing their targeting and biocompatibility, reducing side effects, and increasing their clinical value for patients. Further research and development will help address these issues and drive the application of polymeric micelles in the field of tumor immunotherapy.

## Figures and Tables

**Figure 1 pharmaceutics-15-02622-f001:**
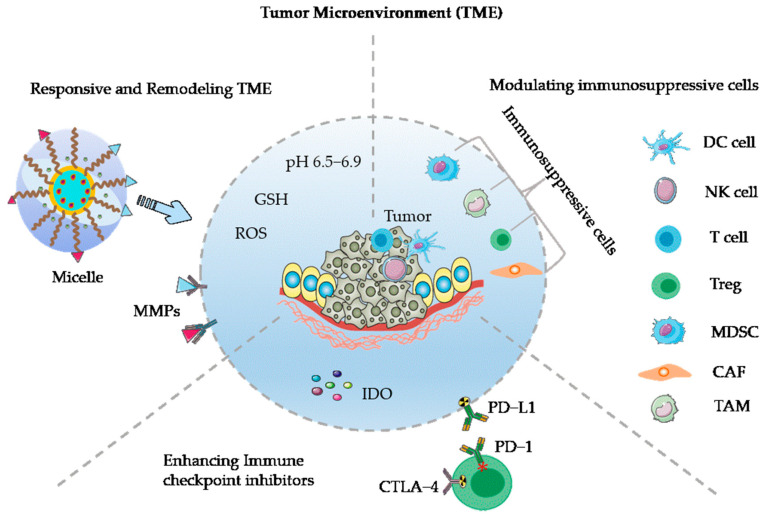
Micelle responsive and remodeling TME, immunosuppressive cells in the TME and ICIs. Micelle responsive TME contain pH, MMPs: Matrix Metalloproteinases, GSH: Glutathione; ROS: Reactive Oxygen Species. Tumor−associated immunosuppressive cells include TAM: MDSC: Myeloid−Derived Suppressor Cell; Tumor−Associated Macrophages; Treg: Regulatory T Cell; CAF: Cancer−Associated Fibroblast. DC Cell: Dendritic Cells; NK cell: Natural Killer Cell; T cell: Thymus Derived cell. ICIs include PD−L1: Programmed Death Ligand 1; IDO: Indoleamine 2,3 Oxygenase; CTLA−4: Cytotoxic T Lymphocyte Antigen−4; PD−1: Programmed Death Receptor 1.

**Figure 2 pharmaceutics-15-02622-f002:**
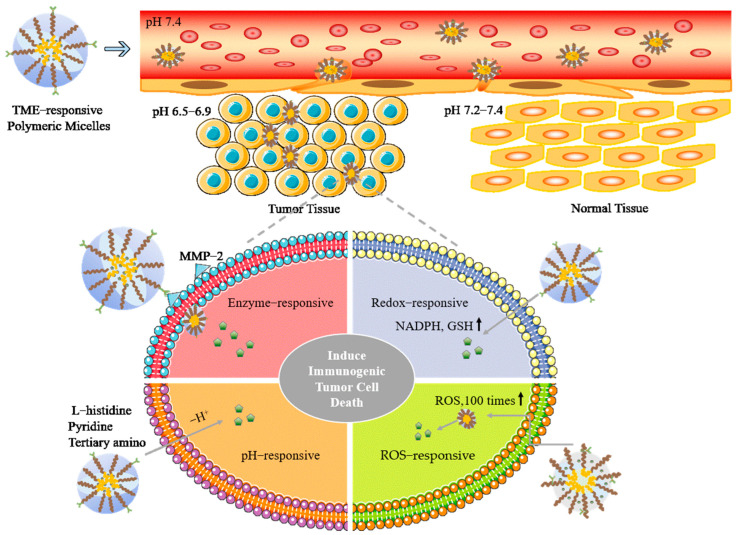
TME−responsive polymeric micelles in tumor immunotherapy. Polymeric micelles achieve targeted drug release to tumor tissues after intravenous administration by responding to a pH range of 6.5−6.9. MMP−2: Matrix Metalloproteinase−2; GSH: Glutathione; ROS: Reactive Oxygen Species.

**Table 1 pharmaceutics-15-02622-t001:** Overview of polymeric micelles for modulate TAMs.

Mechanism of Action	Polymer	Payloads	Tumor Models	Ref.
Targeting TAMs	Phenylboronic acid-poly (ethylene glycol)—poly (ε-caprolactone) and acetylated chondroitin sulfateprotoporphyrin	Imiquimod (R837) and doxorubicin	Breast cancer	[77]
Targeting TAMs	GD modified polycaprolactone-polyethylene glycol	Wortmannin, a specific PI3K inhibitor,	pancreatic cancer	[78]
Targeting TAMs	Hyaluronic acid (HA)-g-poly (histidine) polymeric	Doxorubicin	Breast cancer	[79]
Depleting TAMs	PEG−PLA and Man-PEG-DSPE	Dasatinib	Breast cancer	[80]
Reprogramming TAMs	galactose-functionalized zinc protoporphyrin IX (ZnPP) grafted poly(L-lysine)-b-poly(ethyleneglycol) polypeptide	Poly I:C (PIC, a TLR3 agonist)	Melanoma tumors	[59]
Targeting M2 And Reprogramming TAMs	M2pep targeting peptide modified polyethyleneimin—stearic acid (PEI-SA) and DSPE-PEG	PI3K-γ inhibitor NVP-BEZ 235 and CSF-1R-siRNA	Pancreatic cancer	[81]
Depleting TAMs	DSPE-PEG2000-maleimide	Scrambled MCP-1 peptides	Melanoma	[82]
Depleting TAMs	Dextran-grafted-poly (histidine) copolymer	CSF-1R inhibitor: BLZ945	Breast cancer	[83]
Targeting CD206 of TAM	Quercetin-dithiodipropionic acid-oligomeric hyaluronic acid-mannose-ferulic acid	Curcumin and Baicalin	Lung tumor	[84]
Reprogramming TAMs	PEG-CDM-HES(Hydroxyethyl starch)	Sorafenib and TG100-115	Liver tumor	[85]

**Table 2 pharmaceutics-15-02622-t002:** Summary of polymeric micelles as cancer vaccine platforms.

Mechanism of Action	Polymer	Payloads	Models	Ref.
Accumulation LNs	Amphiphilic poly(L-histidine)–poly(ethylene glycol)	Ovalbumin	C57Bl/6 mice	[126]
Targeting LNs	Dendritic cell membrane/histidine-modified stearic acid-grafted chitosan	Ovalbumin	B16-OVA tumorbearing mice	[48]
Accumulation LNs	Poly(Lhistidine)- poly(ethylene glycol) (PLH-PEG)	Trp2/CpG	B16-F10 tumor-bearing C57BL/6 mice	[127]
Targeting DCs	Polyethylene glycol-phosphatidylethanolamine	OVA_250–264_ peptide	OVA257-264-specific TCR transgenic mice C57BL/6-Tg (TcraTcrb) 1100Mjb/J (OT-I)	[128]
Targeting Skin DCs	Glyceryl monooleate	Hydrophilized melanoma antigen peptide K-TRP-2	Mouse melanoma B16F10 cells and C57/BL6N mice	[129]
Targeting DCs	Mannosemodified poly(ethylene glycol)-block-poly(ε-caprolactone)	Ovalbumin	B16F10-OVA melanoma	[130]
Elevated cytotoxic T lymphocyte	Maleimide-mPEG2000-DSPE-DOPE-MPL	Gp2, HER2/neu-derived peptide	Breast cancer	[131]
Improve DCs activation and enhance antigen-specific T cell responses	Mannosylated block copolymer MAN-P	MHC-I and MHC-II epitopes	B16F10 melanoma	[132]
Target DCs in the Lymph nodes	PEG-PCL using disulfide bond	CpG ODN_1826_, a TLR-9 agonist	B16-OVA and lung metastasis melanoma	[133]

**Table 3 pharmaceutics-15-02622-t003:** Polymer micelles in cancer interventional clinical trials on ClinicalTrials.gov.

Product Name	Drug	Study Status	Conditions	Trial Code
Paclitaxel Micelles for Injection	Paclitaxel	Phase I (Recruiting)	Advanced Solid Tumors	NCT04778839
PPM	Paclitaxel	Phase I (Recruiting)	Non-muscle-invasive Bladder Cancer	NCT05519241
NC-6004	Cisplatin	Phase I/II (Completed)	Pancreatic Neoplasms	NCT02043288
Genexol-PM^®^	Paclitaxel	Phase II (Completed)	Bladder Cancer/Ureter Cancer	NCT01426126
Genexol-PM^®^	Paclitaxel	Phase II (Completed)	Non Small Cell Lung Cancer	NCT01023347
NK105	Paclitaxel	Phase III (Completed)	Breast Cancer Nos Metastatic Recurrent	NCT01644890

## Data Availability

Data are contained within the article.

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
