# Peer review of "Advances in Polymeric Micelles: Responsive and Targeting Approaches for Cancer Immunotherapy in the Tumor Microenvironment"

_pharmaceutics, 2023, doi:10.3390/pharmaceutics15112622_

Round 1
Reviewer 1 Report
Comments and Suggestions for Authors
The authors review the recent advances in the application of the polymeric micelles for the cancer immunotherapy in the tumor microenvironment (TME). Polymeric micelles have emerged as a promising platform for cancer immunotherapy in the TME. In recent years, there have been significant advances in utilizing polymeric micelles to enhance the efficacy of immunotherapeutic approaches and overcome the immunosuppressive nature of the TME. The authors review the main challenges in cancer immunotherapy such as the limited delivery and penetration of therapeutic agents within the TME. Polymeric micelles offer several advantages in this regard. They can encapsulate hydrophobic drugs or antigens within their core, while their hydrophilic shell stabilizes the structure and provides stealth properties. This unique structure allows polymeric micelles to efficiently deliver therapeutic agents to the tumor site and improve their bioavailability. The review covers pH-responsive Polymeric Micelles, Modulating immunosuppressive cells in the TME, Enhancing Immune checkpoint inhibitors (ICIs), Engineering Targeting Polymeric Micelles as Cancer Vaccine Platform, Modulating antigen presentation, Modulating engineered T cells, and Targeting other components of the TME with micelles. The review is well written and accessible for the reader.
I have a few minor suggestions for improvement:
1. Row 220: The abbreviation TME was already introduced. Please remove the repetition.
2. Rows 232-236: the authors should clarify that PD-1 (programmed cell death protein 1) is not a member of the CD28 superfamily. PD-1 is actually a member of the CD28 immunoglobulin superfamily. PD-1 is primarily expressed on activated T cells and B cells, as well as on other immune cells such as natural killer (NK) cells and myeloid cells. Upon activation, PD-1 is upregulated on the surface of these cells and plays a crucial role in immune regulation.
3. The authors can append the review with a notion on the rapidly growing field of the designed cell penetrating peptides (CPP). Protein-derived amino acid sequences provide a rich source of information for the design and engineering of CPPs. Once potential CPP building blocks are identified, they can be synthesized and incorporated into drug delivery nanoassemblies, such as polymeric micelles. The engineering of drug delivery nanoassemblies using predicted CPP building blocks offers several advantages. First, it allows for the design of modular systems where different CPP building blocks can be combined to create nanoassemblies with specific properties and functionalities. Second, by leveraging protein-derived sequences, the resulting nanoassemblies may exhibit enhanced biocompatibility and reduced immunogenicity. A recent paper covering these type of CPP polymeric micelles is “Prediction of Amphiphilic Cell-Penetrating Peptide Building Blocks from Protein-Derived Amino Acid Sequences for Engineering of Drug Delivery Nanoassemblies”, J. Phys. Chem. B 2020, 124, 20, 4069–4078, https://doi.org/10.1021/acs.jpcb.0c01618
Comments on the Quality of English Language
The English language needs only minor editing. No notable issues found.
Reviewer 2 Report
Comments and Suggestions for Authors
The authors have reported that in recent years, there has been notable progress in the field of cancer immunotherapies, significantly advancing the treatment options available for a diverse spectrum of cancer types. Nonetheless, these innovative therapeutic modalities confront a range of challenges when translated into clinical practice. These challenges encompass issues such as the occurrence of elevated immune-mediated toxicity, inadequate accumulation of therapeutic agents within cancerous tissues, and the manifestation of off-target side effects. To surmount these limitations and augment the overall bioavailability of cancer immunotherapies, polymer micelles have emerged as promising solutions. They offer a means to enhance the targeted delivery of therapeutics to the precise site of action, thereby bolstering the efficacy of immunotherapy regimens. This review article is intended to furnish a comprehensive overview of the most recent advances in the realm of micellar-enhanced cancer immunotherapy strategies. These strategies encompass a variety of approaches, including those that involve the responsive and adaptive modification of the tumor microenvironment (TME), the effective modulation of immunosuppressive cell populations within the TME, the augmentation of immune checkpoint inhibitor efficacy, the utilization of cancer vaccine platforms, the refinement of antigen presentation mechanisms, the manipulation of engineered T cells, and the precise targeting of other components within the TME. In concert, the utilization of polymer micelles in these strategies demonstrates substantial promise in the context of tumor immunotherapy, effectively ameliorating the challenges that have traditionally beset conventional cancer immunotherapeutic approaches.
However, the manuscript could benefit from additional characterizations and data to enhance the results and discussion sections. Some of the points with the current manuscript are outlined below, and it is suggested that these be addressed in a revision. Therefore, the reviewer recommends that the paper be reconsidered following these revisions.
The reviewer has the following comments:
Abstract Revision: The abstract should be refined to align with a more scientific style. It is essential to prioritize the discussion of the extensive data encompassed within the manuscript. A concise summary of key findings and their implications should be highlighted prominently.
Figure 1: A schematic cartoon should be reported as Figure 1 and it should feature a unique and innovative infographic that visually represents the key concepts discussed in the manuscript. It is suggested to include polymeric NPs within the infographic.
Table 1: Table 1&2 currently provides only a brief overview and should be expanded to incorporate more comprehensive information, considering the wealth of literature available on this topic. The tables should serve as a valuable reference for readers seeking an in-depth understanding of the subject matter. There is plenty of literature on polymeric nanoparticles (pNPs) for cancer immunotherapy. So, it is recommended to add more data in Tables 1&2. For instance, Seungpyo Hong, Michael Poellman, Kaila and Robert Langer, and many others are focusing on the NPs based formulation for cancer immunotherapy. It would be worth adding their papers to this review article.
Dendrimers-based systems for cancer Immunotherapy: Dendrimers represent a promising frontier in cancer immunotherapy. Their multifaceted properties make them invaluable tools for the development of more effective, precise, and personalized cancer treatment strategies. Research and innovation in this field hold the potential to significantly improve outcomes for cancer patients by harnessing the power of the immune system to combat malignancies. https://doi.org/10.1021/acs.chemmater.2c03705, https://doi.org/10.1021/acs.nanolett.0c00950, https://doi.org/10.1021/acs.nanolett.0c00953.
So, it would be realistic if the authors covered the above topic in the revised manuscript.
Introduction: The introduction section appears to be quite brief and would benefit from a comprehensive revision to help readers clearly understand the scientific problems addressed by this review.
The key Characteristics of pNPs-based Drug Delivery Systems: What are the requirements that a pNPs should fulfill for its application in drug delivery should also be mentioned and explored in the introduction section. For instance, what should be the ideal drug encapsulation, drug loading, drug-releasing, pore size, porosity, morphology, mechanical properties, and other properties?
Future Prospective: A separate section of Future Prospective should be reported.
Commercial pNPs systems: To make the manuscript more comprehensive, commercial pNPs systems used in immunotherapy can be discussed before concluding the manuscript. Thus, a separate section should be added on commercial pNPs systems and the clinical status of different immunotherapies.
Conclusions: In light of the updated data, it is advisable to revise the conclusions to incorporate a more substantial quantitative dataset.
Reviewer 3 Report
Comments and Suggestions for Authors
The manuscript by Cheng et al. summaries the recent advances in the development of polymeric micelles for responsive and targeted approaches for cancer immunotherapy in the tumor microenvironment. This review is timely, well written, and easy to follow. Nevertheless, I have several suggestions that I hope will help the authors improve the manuscript.
- The review lacks a summary of the current status of the development for certain formulations. I think it would be nice to present this information in some way. For example, within the tables, the information could be included indicating the stages of development.
- It is really great that the authors call attention to the limitations of the technology. The authors describe the limitations within the "Conclusion and Future Applications of Polymeric Micelles" section. However, in my opinion, this important information should be presented in a separate section.
- The Figures: The reference to Figure 1 is missing. The legends for both figures are uninformative, especially the abbreviations used are missing. Figure 1 contains punch lines whose meaning is unclear, but which contain schematic receptors.
Author Response
Thank you very much for taking the time to review this manuscript. Please find the detailed responses in the attchment.

Round 2
Reviewer 2 Report
Comments and Suggestions for Authors
All my concerns were answered and I don't have further comments.